# Challenges for Sustaining Measles Elimination: Post-Verification Large-Scale Import-Related Measles Outbreaks in Mongolia and Cambodia, Resulting in the Loss of Measles Elimination Status

**DOI:** 10.3390/vaccines12070821

**Published:** 2024-07-22

**Authors:** José Hagan, Otgonbayar Dashpagma, Ork Vichit, Samnang Chham, Sodbayar Demberelsuren, Varja Grabovac, Shafiqul Hossain, Makiko Iijima, Chung-won Lee, Anuzaya Purevdagva, Kayla Mariano, Roger Evans, Yan Zhang, Yoshihiro Takashima

**Affiliations:** 1World Health Organization, 1211 Geneva, Switzerland; haganj@who.int (J.H.);; 2National Immunization Program, Ministry of Health, Ulaanbaatar 14210, Mongolia; 3National Immunization Program, Ministry of Health, Phnom Penh 12151, Cambodia

**Keywords:** measles elimination, outbreak investigation, vaccine-preventable diseases, Mongolia, Cambodia, Western Pacific Region

## Abstract

The Western Pacific Region’s pursuit of measles elimination has seen significant progress and setbacks. Mongolia and Cambodia were the first two middle-income countries in the Western Pacific to be verified as having eliminated measles by the Western Pacific Regional Verification Commission for Measles and Rubella Elimination, in March 2014 and 2015, respectively. However, both countries experienced large-scale or prolonged importation-related measles outbreaks shortly afterwards, leading to the re-establishment of endemic transmission. We describe the path to initial elimination in both countries and explore these outbreaks’ characteristics, factors contributing to the loss of elimination status, and implications for broader elimination efforts. Data sources include case-based epidemiological and laboratory surveillance reports, historical immunization coverage, genotype data, and published reports of in-depth outbreak investigations. In Mongolia, a single prolonged and large-scale outbreak revealed a hidden immunity gap among young adults and was driven in part by nosocomial transmission, leading to significant morbidity and mortality and loss of elimination status. Cambodia suffered multiple importations from neighboring endemic countries during the global measles resurgence in 2018–2019, complicated by cross-border mobility and significant nosocomial amplification, and the country was ultimately unable to sufficiently distinguish independent chains of transmission, leading to loss of elimination status. Our findings highlight the importance of broadening population immunity assessments beyond children to include adults and specific high-risk groups. Robust routine immunization programs, supplemented by tailored SIAs, are crucial for preventing and managing outbreaks. Additionally, strong outbreak preparedness plans, rapid response strategies, and cross-border collaboration and the global effort to prevent multiple resurgences and large-scale importation-induced outbreaks are vital for maintaining elimination status. The experiences of Mongolia and Cambodia underscore the challenges of sustaining measles elimination in the face of importation risks, shared borders with endemic countries, healthcare system gaps, and population movements. Strengthening the global coordination and synchronization of measles elimination activities is imperative to protect the gains achieved and prevent future setbacks.

## 1. Introduction

In 2003, the World Health Organization (WHO) Regional Committee for the Western Pacific resolved to eliminate measles and concurrently strengthen routine immunization [1] and, in 2005, established 2012 as the target year for measles elimination [2]. In 2010 and 2012, the Regional Committee urged the WHO Regional Director for the Western Pacific and Member States to establish independent national and regional verification mechanisms for measles elimination, including the creation of national verification committees (NVCs) and the Regional Verification Commission (RVC) [3,4].

The verification of measles elimination (defined as the absence of endemic measles virus transmission in a region or country for ≥12 months, in the presence of a well-performing surveillance system) addresses three criteria supported by indicators within the five lines of evidence. These three criteria include the following: (1) documentation of the interruption of endemic measles virus transmission for a period of at least 36 months from the last known endemic case; (2) the presence of verification standard surveillance; (3) genotyping evidence that supports the interruption of endemic measles virus transmission [5]. Mongolia and Cambodia were the first two middle-income countries in the Western Pacific to be verified as having eliminated measles by the RVC; however, both countries experienced large-scale or prolonged importation-related measles outbreaks shortly afterwards. The RVC ultimately determined that endemic transmission had been re-established in Mongolia and Cambodia and the criteria for sustained measles elimination could no longer be demonstrated.

In this manuscript, we describe the characteristics of these measles outbreaks in Mongolia and Cambodia after the verification of measles elimination. We review the factors that led to the outbreaks and to the loss of elimination status and consider implications for the broader measles elimination effort.

## 2. Data Sources

This report references data routinely reported to WHO from Mongolia and Cambodia from case-based epidemiological and laboratory surveillance and historical immunization coverage of routine immunization (via WHO–UNICEF joint estimates of national immunization coverage (WUENIC) [6] and supplemental immunization activities (SIAs) by year of activity and by targeted birth cohort. Genotype data were directly reported to the WHO through the Measles–Rubella Regional Reference Laboratory Network, and accessed through the WHO Measles Virus Nucleotide Surveillance genetic database MeaNS [7]. Additional published data are referenced directly from relevant sources related to in-depth outbreak investigation.

## 3. Mongolia

### 3.1. Path to Measles Elimination in Mongolia

Mongolia is a large and sparsely populated Central Asian country bordering the Russian Federation and the People’s Republic of China. Administratively, it is divided into 21 provinces (“Aimags”) and the capital city, Ulaanbaatar. In the decade following the fall of the Soviet Union in 1990, Mongolia underwent rapid and dramatic social, economic, and political transformations. This period was marked by severe challenges as the country transitioned from a socialist state to a multiparty democratic system and a market economy and experienced sharp drops in foreign aid, causing economic contraction, inflation, and unemployment. Government spending on healthcare significantly reduced, and programmatically the health system shifted from a centrally planned, fully state-funded model to a mixed financing model with the introduction of user fees, leading to a crisis in the provision of health services, including immunization [8]. During this period, intensive urban migration occurred, which among other impacts led to an increase in children living in urban centers but registered for immunization services in their home region and therefore excluded from immunization planning and service delivery.

The measles vaccine was introduced in the national immunization program in 1973, targeting 10–13-month-olds [9], prior to the 1974 World Health Assembly resolution calling for establishment of national immunization programs. The second dose of measles-containing vaccine (MCV) was introduced in 1989 for children 12–18 months, with an updated target age for MCV first dose at 9–12 months. In place of monovalent MCV, the measles–mumps–rubella (MMR) vaccine was introduced in 2009, targeting 9-month-olds for the first dose (MMR1) and 2-year-olds for the second dose (MMR2).

Following the introduction of MCV2 in 1989, there was a decline in measles cases from 1990 to 2002, with periodic relatively small-scale measles outbreaks occurring during 1992 and 1995, and a large outbreak in the capital city in 2001 that affected infants below one year of age and young adults aged 15–24 years [9]. Beginning in 1994, a series of wide age-range supplemental immunization activities were conducted, targeting all birth cohorts born since 1986 at least once and achieving high administrative coverage. Immunization coverage with two doses of measles vaccine has generally been at or above 95% since 2001 [10] (Figure 1). A nationwide serosurvey conducted in 2004 estimated 12–17% measles susceptibility among children who would be aged 13–21 years during the 2015 outbreak [11]. This cohort was partially included in a nationwide SIA conducted in 2007 that achieved 97% administrative coverage, targeting children aged 2–10 years of age (aged 10–18 during the 2015 outbreak).

From 2011 to 2014, no confirmed measles cases were reported in Mongolia. The National Verification Committee of Mongolia submitted documentation that endemic measles transmission had been interrupted for 36 months in the presence of elimination-standard surveillance and convincing evidence of high population immunity from routine and supplemental immunization (Figure 1). In March 2014, Mongolia, together with Australia, Macau, and the Republic of Korea, was in the first cohort of countries to be verified by the RVC to have interrupted endemic measles transmission and achieved measles elimination.

### 3.2. Epidemiology of the Outbreak in Mongolia and Response Measures

Less than one year after being verified to have achieved elimination, in March 2015 it was confirmed that multiple measles cases were simultaneously detected in several districts in Ulaanbaatar City. By June 2015, cases had been reported from all 21 provinces. In total during an outbreak lasting 24 months, a total of 49,185 cases and at least 115 deaths were confirmed from January 2015 to December 2016 (Figure 2). This was the first and still the largest outbreak to occur in a post-elimination setting; the second-largest post-elimination outbreak occurred in Brazil during 2019, with 15,598 confirmed measles cases [12].

This prolonged period of measles virus transmission in Mongolia had two peaks, one during March–August 2015, and a second peak during October 2015–July 2016; the final measles cases were detected in December 2016. Documentation of measles vaccination among cases was very low. The vast majority of cases reported to the WHO had unknown vaccination status; only 2% had documentation of one or two doses. The most affected age groups overall were infants under 2 years and young adults, with a peak at age 18–22 (Figure 3).

The highest attack rate occurred among young adults aged 15–24 years. This age cohort had been understood to be highly vaccinated through a combination of routine immunization and multiple supplemental immunization campaigns, each achieving at least 96% administrative coverage [13] (Table 1). In addition, people born to these birth cohorts had low incidence during the large outbreak in Ulaanbaatar during 2001 [9], suggesting adequate immunity in those exposed to measles during that period. During outbreak investigations, a small percentage (7%) of adults aged 15–24 years had records of vaccination against measles, but in a published case-control study using reported vaccination status by recall, 23% of cases reported they had received at least one dose of measles vaccine in childhood. That study determined that vaccine effectiveness in this group may be as low as 20–70%, which may have contributed to the high attack rate among young adults [13].

Most cases were reported from Ulaanbaatar (33,760, 69%), but all 21 Aimags were affected. During the initial peak, transmission appeared to be most concentrated in the southern Aimag of Umnugovi, bordering China; the most intense transmission then spread north through the central Aimags of Dundgovi, Khentii, Tuv, and the capital, Ulaanbataar. During the second peak, intense transmission was initially more concentrated in western (Bayan-Ulgii, Khovd, Uvs) and northern Aimags (Selenge), but rapidly spread nationwide [14].

In 2015 and 2016 (up to August) a total of 14,462 samples were tested for measles IgM and 6962 (48%) were confirmed positive by laboratory criteria. The genotype of the infection was H1; sequencing data indicated a single importation of the MVs/Hong Kong.CHN/49.12/strain that had been circulating predominantly in China since 2012, which was endemic for measles and had been experiencing a large nationwide outbreak during that time [13]. The outbreak timing coincided with the end of the Tsaagansar Mongolian New Year holiday, during which extensive cross-border travel to China and other countries to visit relatives is common.

This outbreak was characterized by a relatively high morbidity and mortality rate. In total, there were 115 deaths officially attributed to measles, and a published manuscript described a number of severe outcomes including encephalitis or meningitis (38 cases) and pneumonia (334 cases) [14]. The overall case-fatality rate was 0.23%; in the capital city, where the majority of cases were registered, case fatality was low (0.1%), but in many rural Aimags, where medical resources were scarcer, case fatality was near or exceeded 1%, up to 1.4% in Khuvsgul Aimag. Severe outcomes were most commonly reported among infants and adults above 25 years of age [14].

Overall, 33% of measles cases were hospitalized [14], which may have contributed to hospital-associated transmission, which was an epidemiological feature of this outbreak [13,15]. This was an important route of exposure for infants and a driver of mortality during the outbreak. Nosocomial infection was a risk factor for death [14,15]. Most officially reported deaths (111, 97%) occurred during the second wave of the outbreak in 2016; this second peak coincided with a peak of seasonal influenza, and a study suggested that co-infection with other respiratory viruses may have led to the significant increase in mortality, especially among young children during this second phase of the outbreak [15].

In response to the initial cases, Mongolia mounted immediate emergency response measures. These included the following: enhancing measles surveillance and public awareness activities; providing vaccination with measles–mumps–rubella vaccine to close contacts of suspected measles cases; conducting a nationwide non-selective (i.e., regardless vaccination history) outbreak response immunization (ORI) campaign with the measles monovalent vaccine. The nationwide non-selective ORI campaign targeted 371,971 children aged six months to five years of age during May 2015–June 2015, achieving 94% administrative coverage nationwide. Following the ORI, measles incidence decreased to a low level during the early winter months, particularly in young children, until the second peak of transmission beginning early 2016 [14]. During the second outbreak period in 2016, there was a slight decrease in the proportion of cases among children aged 1–6 years (12% in 2016 compared to 15% in 2015), but there was a large increase in transmission among infants not targeted by the immunization response in 2015 and not eligible for routine immunization (18% of cases under age 1 year during 2016 compared to 7% during 2015). After the second outbreak peak, a second ORI was conducted during May 2016, targeting 549,846 adults aged 18–30 years, achieving 88% coverage.

Following the second ORI targeting young adults, measles cases rapidly diminished and the final case was reported in December 2016, marking a total of 22 months of transmission without any epidemiological or genotype evidence to suggest multiple independent importations. In September 2016, the Regional Verification Commission determined that measles transmission had continued uninterrupted for greater than 12 months and that measles elimination could no longer be verified [16].

### 3.3. Root Causes of the Prolonged Outbreak in Mongolia

Mongolia was verified as having eliminated measles in 2014, but subsequently experienced a prolonged period of nationwide measles transmission that led to loss of measles elimination status in 2016. This unexpected event was the result of several factors.

The outbreak revealed a previously unappreciated large immunity gap among young adults aged 15–24 years, born during the highly unstable decade following the fall of the Soviet Union (1990–2000). This “hidden” immunity gap among young adults may have occurred as a result of mass urban migration in the 1990s, potentially concentrating people from remote areas where historical program gaps may have occurred, or leading to an accumulation of unregistered children and missed vaccinations. In addition, published reports suggest the that waning vaccine field effectiveness in this age group may have occurred as a result of compromised vaccination quality or potency during the 1990s, when infrastructure and public services including public health may have been inconsistent [13]. This multifactorial “hidden” immunity gap among young adults was an unrecognized significant threat to the sustainability of measles elimination. Despite serological evidence of an adult immunity gap in 2004 that was not targeted by the nationwide SIA in 2007, the scale of this gap was not fully understood, and its potential impact was not heavily weighted by the National Verification Committee or the Regional Verification Commission in their determination that Mongolia had achieved measles elimination in 2014. The role of this immunity gap in driving transmission was also not addressed during the initial outbreak response immunization campaign in May–June 2015, which was high quality and achieved high coverage but followed a “traditional” measles outbreak response strategy and targeted young children aged 6 months to 5 years. This led to a further shift of measles epidemiology towards adults and infants below the age of vaccination and wider geographic dissemination of the virus, and this set the conditions for the second wave of transmission during the first quarter of 2016.

Hospital-associated transmission was another major contributor to the scale and duration of the outbreak, particularly at the beginning, when multiple local outbreaks simultaneously occurred in Ulaanbaatar City during the second quarter of 2015, and weak hospital infection prevention and control practices related to measles and other respiratory viruses was another important threat to the sustainability of measles elimination in Mongolia. Healthcare exposure during the incubation period was a significant risk factor for measles infection [13] and measles death [14,15]. Improved infection prevention measures could have prevented the amplification of measles transmission in the healthcare setting, particularly in the early stages of the outbreak, and potentially could have limited the extent of the outbreak after importation. Initial cases were simultaneously detected in several district hospitals before being detected in Ulaanbaatar, suggesting that low-level transmission had already been occurring before the outbreak was detected and that hospital-associated amplification of transmission after importation likely was an important factor in the early spread of the outbreak.

## 4. Cambodia

### 4.1. Path to Measles Elimination in Cambodia

Cambodia is a lower middle-income country in the Greater Mekong Delta, divided into 25 Provinces, including the Phnom Penh Municipality, and 103 “operational districts” (ODs) (during 2016–2019, there were between 98 and 100 ODs). Cambodia shares porous land borders with three countries that remain endemic for measles: Thailand, Lao People’s Democratic Republic, and Vietnam. There is significant mobility across these borders, resulting in distinct challenges for immunization planning and disease surveillance. Several cities in Cambodia are major international tourist destinations, drawing 6.2 million visitors from within the region and beyond in 2018 [17]. Within the Greater Mekong Delta, which includes these neighboring countries, along with Yunnan Province of China and Myanmar, there are an estimated 3–5 million migrant workers supporting generally low-skilled work on an irregular, seasonal, or short-term basis. High-traffic crossing points into Cambodia, including Poipet on the Thai border, Bavet on the Vietnamese border, and Tropaeng Kreal on the Lao PDR border, among many others, are important routes for the potential importation of the measles virus. In addition, there is very easy movement across borders in some parts of the country from village to village, and these countries share a number of cross-border populations and ethnic groups maintaining family ties across national boundaries, including the Khmer Krom, Hmong, and various hill tribes such as Lamam, Brao, Kravet, Jarai, and Chong. In some cases, these populations may lack a sense of specific national identity, remain unregistered in any country, and maintain a highly mobile style of living, creating conditions that lead to the virus being systematically missed by both routine and supplemental immunization planning and implementation.

Following the end of the Khmer Rouge period in 1979, Cambodia underwent significant political and economic transformations, including rebuilding a health sector that had been devastated by the dismantling of medical facilities, systematic targeting of skilled healthcare workers and national resources, and policies focused nearly exclusively on agricultural labor. The government of Cambodia established the expanded program on immunization (EPI) in 1986, and program activities reached all provinces of the country by 1988. The Cambodian Ministry of Health (MOH) established the National Immunization Program (NIP) in 2000. The measles monovalent vaccine was introduced into routine immunization in Cambodia in 1986 at 9 months of age. Reported coverage of the first dose of measles-containing vaccine (MCV1) has trended upward since 2004, with consistently high coverage (over 90%) since 2009 and reaching 95% in 2015 (Figure 4). In 2015, out of 25 provinces in the country, 13 provinces reported >100% (100–122%) MCV1 coverage, while 10 provinces reported <90% (73–89%) (Figure 5). The Cambodia MOH introduced the second routine dose of MCV at 18 months of age in late 2012. The first routine dose of MCV given at 9 months of age was replaced with the measles–rubella vaccine (MR) in late 2013; MCV2 was switched to MR during 2015. Coverage with MCV2 slowly progressed from 63% in 2013 to 72% in 2015, with only two provinces (8%) in 2013 and one province (4%) in 2015 meeting the 95% coverage target.

Cambodia conducted six large-scale mass vaccination campaigns with MCV from 2000 to 2014, achieving very high reported coverage (Figure 4). A nationwide measles catch-up SIA, targeting 4.5 million children aged 9 months to 14 years, achieved 98% and 103% during two different phases between October 2001 and April 2003. In 2005, high-risk areas were selected, comprising approximately 10% of the total population, such as mobile Vietnamese villages and boat dwellers along the Tonlesap river and lake, Muslim communities, Khmer and Vietnamese slum areas in Phnom Penh, and migrating Khmer groups at the Thailand border for a mass vaccination campaign targeting 134,067 children aged 9 to 59 months and achieving 86% coverage. During 2007, a nationwide follow-up measles SIA targeted 1,457,235 children from 9 to 59 months of age, achieving 105% coverage. In 2011, a nationwide follow-up measles SIA achieved 105% coverage of 1.5 million children aged 9 months to 59 months. In 2013–2014, a wide age range nationwide SIA targeted 4,345,392 children aged 9 months to 14 years, achieving 105% coverage. This was followed in 2014 by selective catch-up activities in high-risk communities in 23 provinces, vaccinating 11,115 children with MR1 and 21,158 children with MR2.

Measles case-based surveillance was started in 1999. In 2014, a broader case definition for suspected measles cases was updated to include fever and maculopapular rash. The last laboratory confirmed case was detected in November 2011; in March 2015, the Regional Verification Commission for Measles Elimination in the Western Pacific (RVC) verified that Cambodia had interrupted endemic measles transmission for a period of at least 36 months since 2011. Beginning January 2016, Cambodia experienced a series of importations of measles virus that led to a prolonged outbreak until June 2017, and from December 2018 to December 2020 (Figure 6).

### 4.2. Outbreaks, Root Causes and Loss of Elimination Status in Cambodia

#### 4.2.1. January 2016–June 2017

Beginning in January 2016, less than one year after achieving measles elimination, Cambodia began detecting laboratory-confirmed cases of measles. In total during January 2016–June 2017, a total of 66 lab-confirmed cases were reported from 33 operational districts in 16 provinces (Figure 5). There were no reported deaths.

A robust outbreak investigation and response was initiated, including local outbreak response immunization, nationwide inclusion of a zero-dose MR vaccine to children aged 6 months–<9 months and an extra vaccination opportunity offered to children aged 24 months to <59 months, and catch-up vaccination in affected ODs. A scheduled Japanese Encephalitis immunization campaign was broadened to include the MR vaccine, non-selectively targeting children aged 9 months to 59 months in 11 provinces and achieving 91% coverage. Finally, a nationwide preventive supplemental immunization campaign that had been previously scheduled for October 2017 was conducted early, during March–April 2017, achieving 90% administrative coverage, but post-survey coverage was 75%. Mop-up activities targeted areas with lower coverage and delivered an additional 103,056 doses. Vitamin A and Mebendazole were also provided as part of this activity. The last case of lab-confirmed measles in 2017 occurred on 8 June. Intensive case investigation and case finding determined that 3 cases were classified as imported, 54 were import-related, and 9 were unknown. During the 17-month duration of the outbreak, there were four gaps longer than 30 days, despite active case finding and surveillance meeting key performance indicators for sensitivity. In two instances, the gap was followed by new importations identified by case investigation; in two instances, the gap was interrupted by cases with no epidemiological link, no travel history, and an unknown source.

There were two genotypes detected during the 2016–2017 outbreak: D8 and B3. B3 had never been detected in Cambodia before 2016, but the B3 genotype detected during this outbreak also had no identical match in MeaNS [7]. Among the B3 and unknown genotype cases, 7 cases had unknown source of infection; 15 cases had contact with a hospital, there were 37 secondary cases in five clusters, and 4 cases that could not be assigned to a cluster. There were three clusters of D8 cases and transmission chains: two unlinked cases in different provinces, with a history of travel to Thailand; one case from Phnom Penh with no history of travel and contact with a hospital during the incubation period; and a cluster of four cases linked to history of travel to Thailand. Genetic sequencing data supported repeated importation from Thailand as the likely source.

To guide RVC deliberations, the WHO conducted an analysis to reassess discarded cases to determine whether true measles cases potentially testing false negative for measles IgM due to timing of specimen collection might have occurred during the gaps in transmission that might indicate uninterrupted transmission exceeding 12 months. The analysis found a total of 342 IgM-negative discarded cases where the specimen was collected <4 and >28 days after rash onset. Of these, 221 had a documented history of vaccination, leaving 121 cases potentially erroneously discarded. All 121 had at least one of the classic symptoms of measles (cough, coryza, conjunctivitis). Cases were reviewed in detail to identify “probable” false negative cases by epidemiological or clinical criteria, yielding 30 such cases: 6 cases had contact with a confirmed measles case during the incubation period, 23 cases had Koplik’s sign (a clinical finding highly associated with measles infection) and were from an OD where confirmed cases had been reported, and 1 case met all these criteria. Including these additional probable false negative cases indicated that the maximum duration of continuous transmission may have been 47 weeks, including a 35-day gap in transmission during weeks 37–39, 2017 (Figure 7).

The RVC determined that the data indicated that the 2016–2017 outbreak was the result of multiple independent importations and low-level transmission sustained largely by nosocomial transmission. Due to aggressive outbreak response and intensive case investigation, the outbreak was contained with a maximum duration of uninterrupted transmission of less than 12 months, and the RVC verified that measles elimination had been sustained.

#### 4.2.2. December 2018–December 2020

During December 2018–December 2020, Cambodia experienced a second, larger outbreak after achieving elimination. A total of 1056 confirmed measles cases and no deaths were reported to the WHO during this period. Most cases were reported from major reference hospitals throughout Cambodia; only 12% of cases were reported from community health facilities. The majority of cases (835, 79%) were among children aged 6 months to 10 years, but 7% of cases occurred among infants below the age of eligibility for MCV1 at 9 months. Most cases were unvaccinated (723, 68%) or had only one dose (172, 16%).

Initially in 2019, the outbreak was concentrated in Siem Reap district, Cambodia’s most important area for national and international tourism and the location of a major children’s referral hospital. Overall, the capital, Phnom Penh, was the most affected (23% of cases), but there was widespread geographic distribution; cases were reported from 24 of 25 (96%) provinces and 77 of 103 (75%) ODs. During this outbreak, B3 and D8 genotypes were again detected. Detailed genetic characterization of the B3 viruses, detected from Siem Reap OD, indicated one distinct sequence identification (DSID 5937). According to data in MeaNS, globally this DSID has only been identified once before in the Philippines, one month (January 2019) before first detection in Cambodia [7]. The rash onset dates of the B3 cases ranged over 23 weeks from 18 February until 27 July 2019. No cases selected for testing with rash onset after 21 November 2019 resulted in a B3 genotype. It is highly likely that the B3 transmission within Cambodia ended in late July or August 2019. Intensive case investigation indicated multiple independent chains of transmission of the same B3 measles strain, resulting from multiple importations to this major international tourism center. Genotype lineage D8-2019B was the most common in 2019 and 2020. Epidemiological investigation identified at least three independent importations from Vietnam and at least eight independent importations from Thailand. There were four other lineage clusters epidemiologically linked to repeated introductions into the country and 13 other genotype lineages with sporadic detection during the outbreak period.

Epidemiological investigation indicated that nosocomial transmission played a much more important role in the spread of this outbreak and wide geographic distribution than community-based transmission. Medical care in Cambodia is often sought in referral hospitals at great distances from home. From a subset of 1047 cases reported to the WHO between December 2018 and October 2020, 854 (82%) cases were admitted at a hospital for treatment before, during, or after rash onset while symptomatic. Of cases without direct epidemiological linkage to a confirmed case, 58% reported healthcare facility exposure during the incubation period. Of 145 cases with probable nosocomial infection during this period, 125 (86%) exposures occurred in one of five major referral children’s hospitals. Table 2 illustrates the large number of measles cases admitted for care in these hospitals, and that cases linked to these hospitals were detected across a wide geographical area within Cambodia. Figure 8 visualizes the exposure patterns of a subset of these cases. Of 590 individual cases with available contact-tracing data during 1 January 2019 to 29 December 2019, 579 (98%) reported hospitalization during the symptomatic period, 139 (24%) reported they had been exposed to a measles case in a hospital prior to symptom onset, and 81 cases (14%) reported exposure to a symptomatic measles case outside of the hospital setting. Of these 81 cases, 38 (47%) reported person–person exposure to contacts who had themselves been previously exposed to measles in a hospital. The investigation determined that, despite efforts to strengthen hospital infection prevention and control measures, healthcare-acquired infections were propagating continuous measles virus transmission and seeding measles virus across a wide geographic area through patients returning to their home province after infection during hospitalization.

Following initial detection of confirmed measles cases in December 2018, Cambodia mounted an urgent outbreak response, including intensive case investigation and contact tracing, sensitization of local health authorities, rapid routine coverage assessment at the village level (Figure 9), and local outbreak response immunization, which vaccinated a total of 158,189 children aged 6 months to less than 15 years. A nationwide catch-up campaign vaccinated an additional 102,804 children aged 6 months to 15 years, and in Siem Reap district, a sub-national non-selective campaign vaccinated 96,893 children aged 6 months to 15 years during April–May 2019, achieving 89.3% coverage.

The endemicity of measles virus transmission in both Thailand and Vietnam and the pressure of imported viruses from these countries was low from 2015 to the first half of 2018, but significantly increased in Thailand from the second half of 2018 to the first quarter of 2020 (D8 and B3) and in Vietnam from early 2018 to the first quarter of 2020 (D8). This resulted in the increased importation of measles into Cambodia from early 2019 to the first quarter of 2020. Epidemiological case investigation and genotyping sequence data indicated numerous independent measles importations of predominantly the D8 virus during this prolonged period of transmission (Figure 10). As case counts increased and public health resources became strained towards the end of 2019, the capacity for intensive case investigation and genotyping to identify independent chains of importation was reduced. From October 2019, most lab-confirmed cases lacked information on epidemiological linkage, and most cases lacked serological samples for laboratory case confirmation or virological samples to facilitate genotyping; as a result, until the end of the outbreak, Cambodia was unable to differentiate independent importations or unlinked chains of transmission.

During its 2021 meeting, the RVC concluded that “[a] D8 lineage, detected between February 2019 and April 2020, has a 15-month period of likely transmission that included many sporadic measles cases of unknown genotype and origin” and determined that measles elimination status had not been sustained.

## 5. Conclusions and Implications for the Global Measles Elimination Effort

The achievement of measles elimination in Mongolia and Cambodia illustrated the success that is possible by a strong national commitment to the regional public health goal and strong immunization program focused on a national goal. The path to achieving elimination in these countries reflected decades of intensive investment by the countries and partners to achieve the highest possible vaccination coverage among children through routine immunization program and supplementary immunization activities targeting young birth cohorts, and to implement high-quality, sensitive, and representative surveillance. The prolonged outbreaks in these countries following the importation of the virus from neighboring endemic countries, which led to loss of elimination so soon after this impressive achievement, revealed vulnerabilities that had been previously unrecognized during the lead-up to achieving elimination.

The experiences of Mongolia and Cambodia provide several important lessons for the global initiative to eliminate measles from all six WHO regions. First, these events underscore the need for National Verification Committees (NVCs) and Regional Verification Commissions (RVCs) to consider additional issues when assessing the elimination status of countries, with particular attention on the “sustainability” criteria. First, these bodies should extend their assessment and evaluation of population immunity beyond the traditional focus on children. It is essential that these bodies consider a wider age demographic, including adults, as well as high-risk groups within the population in their efforts to understand the risk of large outbreaks after future importation, and sustainability of measles elimination. NVCs should collaborate closely with National Immunization Programs (NIPs) to identify subpopulations with significant immunity gaps who may be left out of routine immunization programming, such as young adults and migrant workers. These are often the segments of the population that fuel outbreaks due to their mobility and concentration in certain employment sectors. Collaboration of the NIP with non-traditional sectors such as the Ministries of Labor or Immigration may be required in order to develop strategies for the vaccination of groups normally excluded from routine immunization programming. Proactive identification and vaccination of these groups can significantly reduce the risk of outbreaks. In addition, this may include advocating for countries to take steps to address threats to sustainability that are outside the immunization program, such as poor hospital infection prevention and control practices, and cross-border movement.

Countries that have interrupted endemic transmission and achieved measles elimination status continue to be vulnerable to the importation of the measles virus from other endemic countries, particularly if they share porous borders with these countries, and be at risk for losing elimination status if the imported virus finds a significant population immunity gap, leading to prolonged transmission. The large outbreak in Mongolia is one example of the risk that, even in countries with high estimated vaccine coverage among children, significant but unknown immunity gaps might exist or emerge in the population. This may result from undocumented or under-appreciated historical disruptions to the immunization program (e.g., cold chain failures) and other events such as mass population movement or temporary immigration such as labor migration, that might impact coverage estimates or immunization program planning. Countries at particular risk of frequent importation of the measles virus from neighboring endemic countries, as exemplified by the experience of Cambodia during 2016–2020, may have the additional burden of needing to invest significant resources and effort into both epidemiological investigation and the collection of virological samples for genotyping in order to distinguish independent importations and limited chains of transmission from continuous re-establishment of endemic measles transmission. Differentiating endemic transmission from frequent importation–elimination cycles will remain difficult while there is endemic transmission across porous borders and state-less populations are accounted for in no country’s denominator.

Cambodia’s experience in particular highlights that hospital-associated amplification of measles transmission is a serious potential threat to countries working to achieve or sustain measles elimination. During the 2019–2020 outbreak in Cambodia, an overwhelming majority of cases either visited healthcare facilities during their symptomatic period or were exposed to hospital facilities during the incubation period, and confirmed exposures in a small number of referral hospitals were linked to cases with widespread geographic distribution within the country. This experience illustrates the importance of strengthening infection prevention and control practices within healthcare settings (including strict isolation of measles cases in hospitals and clinics where possible), to prevent such facilities from becoming epicenters for disease spread, and is a reminder that sustaining elimination requires investment in health system resilience, inter-sectoral collaboration, and community engagement.

Strong political support for and a proactive attitude towards prevention and preparedness are therefore required in order to sustain elimination status. Strong routine immunization programs that aim to ensure equitable delivery of immunization through strategic and high-quality micro-planning, outreach, and catch-up vaccinations should be the backbone of outbreak prevention. Outbreaks and individual cases should be carefully investigated to identify root causes for lack of vaccination and risks for virus importation. This use of measles cases as a “tracer” may identify systematic issues related to routine immunization strategy and implementation, can guide broader immunization systems strengthening, and focus surveillance efforts. Supplemental immunization activities (SIA) should be implemented to close immunity gaps before an imported case leads to a large outbreak and potential loss of elimination status. SIAs, tailored to the needs of specific at-risk groups, are vital for maintaining the hard-won gains against measles. Development of and regularly updating national outbreak preparedness plans is critically important to ensure that the impact of potential outbreaks can be minimized. This includes meticulous and timely investigation of measles cases and outbreaks to identify and disrupt all chains of virus transmission through enhanced surveillance and contact tracing, and to understand the transmission dynamics and root causes of outbreaks to guide both the immediate response as well as programmatic changes or strengthening.

Stronger coordination and synchronized operations should be established and promoted at the global level to achieve and sustain measles elimination. Cambodia and Mongolia invested significant effort and resources towards achieving measles elimination, but they had neighboring countries still experiencing large-scale endemic transmission with frequent cross-border importation of the virus. The WHO and the global community must intensify their promotion of coordinated and synchronized activities towards measles elimination. The vulnerability of countries (particularly lower- and middle-income countries) who have achieved measles elimination to repeated importation of the measles virus from endemic countries highlights the need for greater collaboration and coordination across neighboring countries and regions. This includes the sharing of best practices, alignment of messaging to shared special populations, sharing of surveillance and epidemic intelligence data in a systematic and standardized manner between countries and across regions, and synchronization of activities targeting highly mobile groups that may span borders and which may not otherwise be included in routine national planning. This synergy is essential for the efficient use of resources and for effectively managing cross-border disease threats.

## 6. Conclusions

These experiences of Mongolia and Cambodia demonstrate that the asynchronous and uncoordinated pursuit of measles elimination in individual countries can ultimately create a vulnerability in those countries that do invest effort and resources towards achieving that goal. While countries and regions have made great progress towards the goal of measles elimination, without a global measles eradication goal and accelerated efforts coordinated at the regional and global level some of these successes have become Pyrrhic victories. Countries that have made substantial progress towards or have reached elimination should not be penalized by their successes by the burden of preventing re-establishment in the face of the continued regional and global circulation of measles. A global measles eradication goal is not merely aspirational but would be an essential step in safeguarding the investments and public health triumphs of countries that have achieved elimination status, and ultimately lead to a legacy of enhanced global health security and a healthier world.

## Figures and Tables

**Figure 1 vaccines-12-00821-f001:**
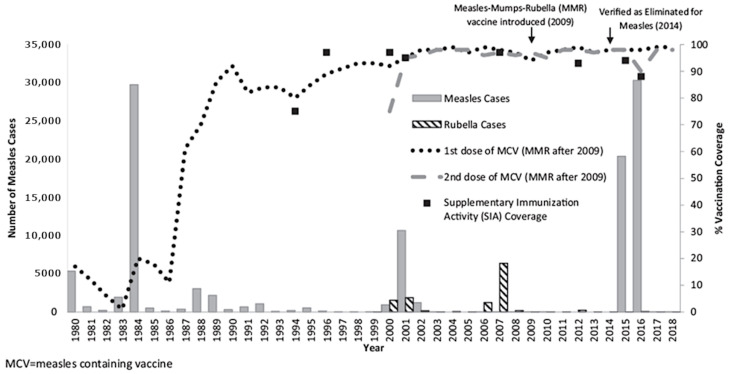
Measles and rubella morbidity and measles-containing vaccination coverage by year—Mongolia, 1980–2018. Reprinted from Ref. [10] with permission from Elsevier.

**Figure 2 vaccines-12-00821-f002:**
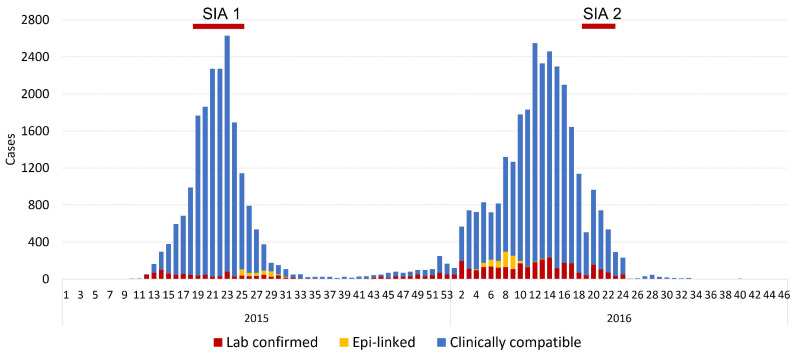
Epidemic curve of confirmed measles cases by epidemic week of rash onset—Mongolia, January 2015–December 2016. Data source: case-based measles surveillance data officially reported to the WHO. N = 49,185 cases.

**Figure 3 vaccines-12-00821-f003:**
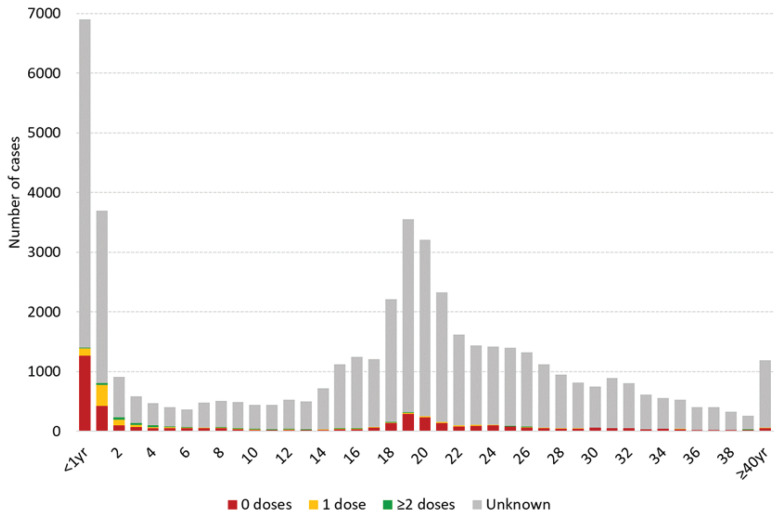
Age distribution of confirmed measles cases and reported immunization status—Mongolia, January 2015–December 2016. Data source: case-based measles surveillance data officially reported to the WHO. N = 49,185.

**Figure 4 vaccines-12-00821-f004:**
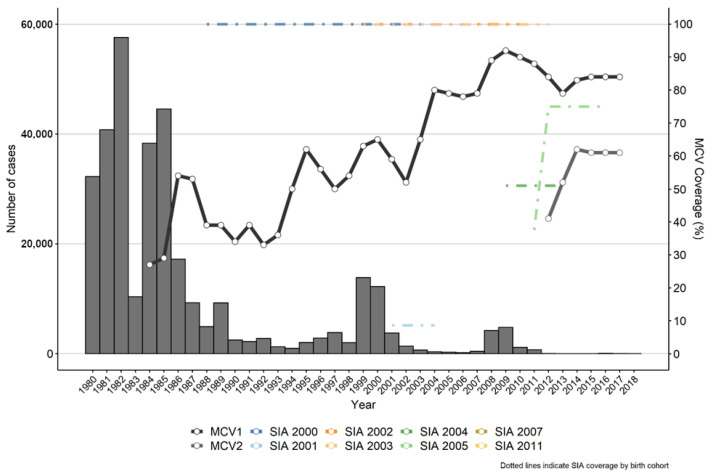
Measles morbidity and measles-containing vaccination coverage by year—Cambodia, 1980–2018. Dotted lines indicate SIA coverage by birth year. Data source: case-based surveillance data reported to WHO, WHO-UNICEF Estimates of National Immunization Coverage, and national estimates of SIA coverage by birth cohort, from Cambodia MOH reported to the WHO.

**Figure 5 vaccines-12-00821-f005:**
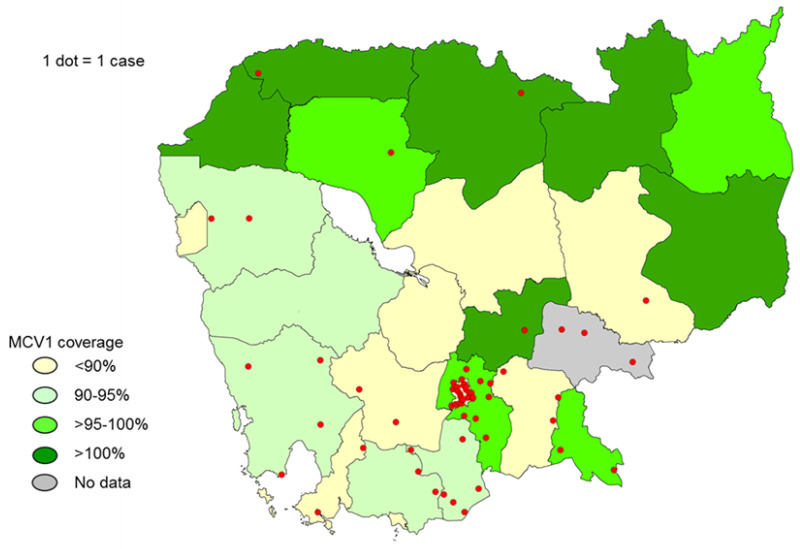
Geographic distribution of confirmed measles cases—Cambodia, 2016–2017, and reported Provincial MCV1 coverage in 2015. Source: Case-based surveillance data and immunization coverage data reported to the WHO.

**Figure 6 vaccines-12-00821-f006:**
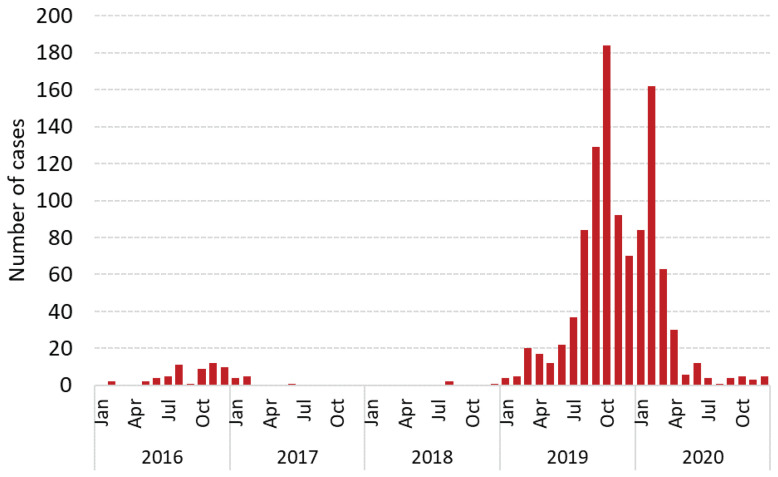
Epidemic curve of laboratory-confirmed measles cases by month of rash onset—Cambodia, 2016–2020. Data source: case-based measles surveillance data officially reported to the WHO. N = 1056 cases.

**Figure 7 vaccines-12-00821-f007:**
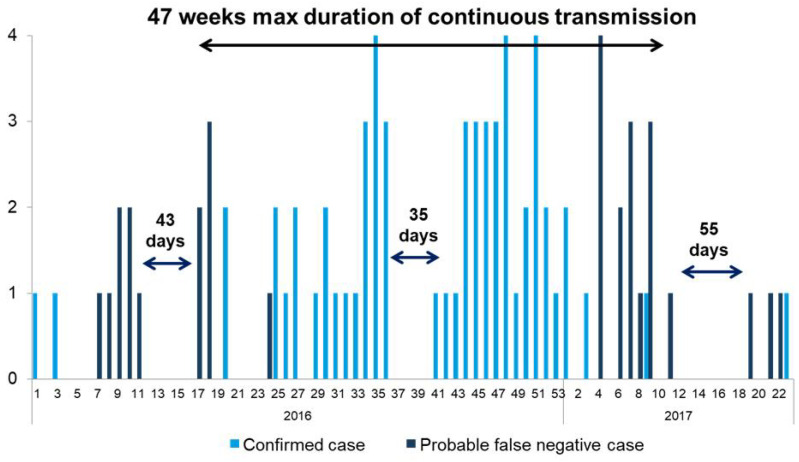
Epidemic curve of confirmed (n = 66) and probable (n = 30) measles cases—Cambodia, 2016–2017. “Probable false-negative cases” include cases that were not lab-confirmed but that had epidemiological or clinical characteristics compatible with a potential false negative laboratory result. Source: Case-based surveillance data reported to the WHO.

**Figure 8 vaccines-12-00821-f008:**
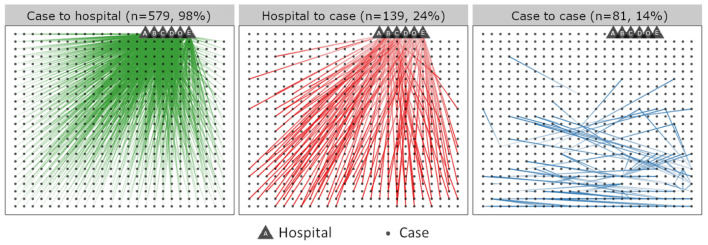
Contact tracing indicating the role of nosocomial transmission in the amplification of measles spread—Cambodia, January–December 2019. The dots in each panel represent N = 590 individual cases for which contact-tracing information was available, epidemiologically linking each case to one or more cases or to exposure to a healthcare facility, or who visited a healthcare facility while symptomatic. The same cases are represented in each panel. Each triangle represents a different hospital indicated by letters A–E. Hospital “O” represents any healthcare facility besides the five largest referral hospitals in Cambodia. Green lines indicate cases that were hospitalized during the infectious period (n = 579, 98%). Red lines indicate cases with documented exposure to a measles case at a hospital (n = 139, 24%). Blue lines indicate epidemiological linkage between individual cases (n = 81, 14%).

**Figure 9 vaccines-12-00821-f009:**
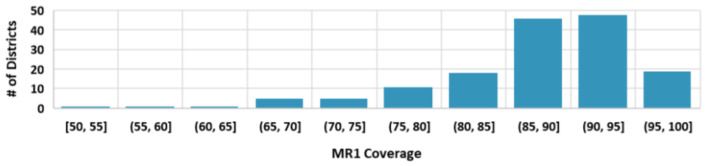
Histogram of average village-level routine MR1 coverage by district from rapid coverage assessments during case investigation—Cambodia, 2019–2020.

**Figure 10 vaccines-12-00821-f010:**
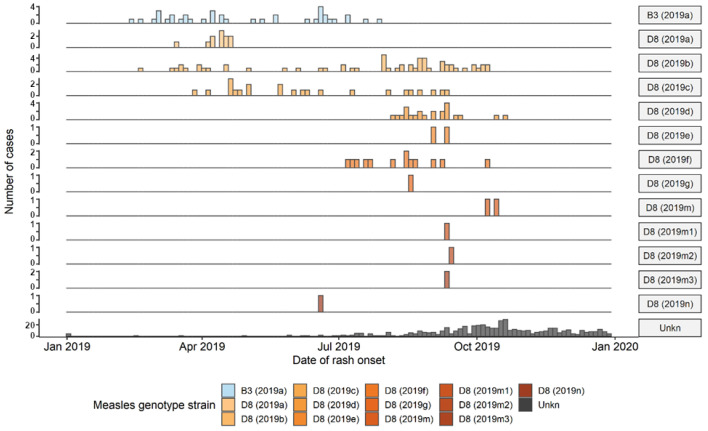
Epidemic curve by genotype strain and date of rash onset—Cambodia, January 2019–December 2019. Source: Case-based surveillance and laboratory data officially reported to the WHO.

**Table 1 vaccines-12-00821-t001:** Supplementary immunization activities (SIA) using measles-containing vaccine—Mongolia, 1994–2016 *.

Year of SIA	Vaccine Used	Location	Age Group Targeted	Reported Administrative Coverage (%)
1994	Measles	National	3–7 years	75
1996	Measles	National	9 months–11 years	97
2000	Measles	National	9 months–7 years	97
2001	Measles–Rubella	Ulaanbaatar	6 months–30 years	95
2007	Measles	National	2–10 years	97
2012	Measles–Rubella	National	3–14 years	93
2015	Measles	National	6 months–<6 years	94
2016	Measles–Rubella	National	18–<30 years	88

* Reprinted from Ref. [10] with permission from Elsevier.

**Table 2 vaccines-12-00821-t002:** Cases linked to healthcare exposure—Cambodia, December 2018–October 2020. Source: case-based surveillance data reported to the WHO.

Hospital	Cases Admitted to Hospital (# of Unique Provinces)	Cases Linked to Hospital Exposure	Provinces with Cases Linked to Hospital Exposure	Districts with Cases Linked to Hospital Exposure
A	290 (21)	51	9	27
B	71 (16)	22	8	12
C	40 (1)	28	4	8
D	64 (8)	21	4	10
E	13 (2)	3	1	3
Other HCF	101 (19)	20	11	17

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
