# Peer review of "Challenges for Sustaining Measles Elimination: Post-Verification Large-Scale Import-Related Measles Outbreaks in Mongolia and Cambodia, Resulting in the Loss of Measles Elimination Status"

_vaccines, 2024, doi:10.3390/vaccines12070821_

Round 1

Reviewer 1 Report

Comments and Suggestions for Authors

This manuscript is well organized and the review by the authors of changes within countries that led to discrepancies in the health system and immunization is beneficial information to include when exploring sustainability of measles elimination. Understanding this background is important to identifying how immunization gaps develop and the authors do a nice job covering this wide range of information in a cohesive and concise manner.

Mongolia evaluation comments:

Can there be more noted into what evidence was used to demonstrate high population immunity? (line 111-112) Given that this gap of information was clearly a blind spot that wasn’t seen when elimination verification was reviewed. Can the authors expound on why the immunity gap in young adults was not visible to the RVC though convincing serological evidence was presented?

Figure 2 it would be recommended to make the axis text smaller to allow better accommodation and readability of the epi week.

Line 130-132 can the authors break down the age range of the two outbreak peaks in a more specific manner, given that it was primarily the under 1 and 1 yo and those that were 18-22 that had most cases for the outbreak. The 1-24 age breakdown really doesn’t fit for those 2-14 compared to those over 24 excluded in that age breakdown of the main age groups driving the outbreaks.

Is there data that can be provided by the author in regard to the age breakdown of cases in the second peak? It is noted that after the first SIA that transmission in younger ages decreased but is data available to demonstrate the age differences in the two peaks of the outbreak if there were any. This data will be additionally helpful to identify the use or limitation in use of administrative coverage, which should be noted more clearly as a limitation in evaluation of immunity.

Cambodia evaluation comments:

Figure 7 legend doesn’t appear to go with figure.

Figure 8 while the attempt to depict the connection between nosocomial infection and transmission is appreciated, it is unclear if this is the best depiction of this phenomenon. It is unclear in the text if the dots are different individuals in the green and red figures. The assumption is that because lines connect the same dots in green and red panels, however the blue grid would indicate that these are the same if transmission can be linked across the dots. If the dots are the same, then it should be clarified if these are only nosocomial cases and not any additional connected cases in the community or the percentage of the cases in the red that are community acquired vs. additional hospitalized individuals should be clarified. Is there a way in the patient-to-patient transmission to be color coded to identify dots that were green so in the hospital during infectious period and red those linked to those cases. A greater description of this figure would be appreciated by the reader to aid in understanding.

Figure 9 is this data prior to the outbreak or from supplementary immunization campaigns. Clarifying this point would help in understanding this figure.

Comments on the Quality of English Language

Line 43 in the introduction should be “The three criteria include…”

Line 285 targeted used an extra time in the sentence.

Line 389 there is a period after onset that does not include the fragment of the sentence “while symptomatic”.

Line 459 should be “…and cross-border movement.”

Line 490 proactive does not need to be hyphenated.

Line 495 route should be root for “root cause”

Author Response

Dear referee,

Thank you for the thoughtful review of our manuscript. We have fully addressed each of the identified issues and believe this has strengthened and clarified the manuscript. Your specific comments are listed below, followed by our response in red text. We have included a revised PDF and a latex script including tracked changes.

Mongolia evaluation comments:

Can there be more noted into what evidence was used to demonstrate high population immunity? (line 111-112) Given that this gap of information was clearly a blind spot that wasn’t seen when elimination verification was reviewed. Can the authors expound on why the immunity gap in young adults was not visible to the RVC though convincing serological evidence was presented?

---A reference to figure 1 was added, showing the historical vaccination coverage through routine and supplemental immunization. We have also edited the discussion to recognize that there was serological evidence of an unaddressed immunity gap among adults but that the risk to elimination was not fully considered.

 Figure 2 it would be recommended to make the axis text smaller to allow better accommodation and readability of the epi week.

--We have revised the figure axis text, thank you

Line 130-132 can the authors break down the age range of the two outbreak peaks in a more specific manner, given that it was primarily the under 1 and 1 yo and those that were 18-22 that had most cases for the outbreak. The 1-24 age breakdown really doesn’t fit for those 2-14 compared to those over 24 excluded in that age breakdown of the main age groups driving the outbreaks.

--We have revised the description of the age distribution to more specifically indicate that the most impacted age groups were aged under 2y and those aged 18-22.

Is there data that can be provided by the author in regard to the age breakdown of cases in the second peak? It is noted that after the first SIA that transmission in younger ages decreased but is data available to demonstrate the age differences in the two peaks of the outbreak if there were any. This data will be additionally helpful to identify the use or limitation in use of administrative coverage, which should be noted more clearly as a limitation in evaluation of immunity.

--Transmission during the first peak decreased somewhat among young children but during the second peak there was a marked increase in attack rate among young infants not eligible for routine vaccination and not targeted by the SIA. We have added more details to the text to indicate the slight difference in 2016 in transmission among the approximate age group targeted in the 2015 SIA, but the great increase in transmission among young infants.

 Cambodia evaluation comments:

Figure 7 legend doesn’t appear to go with figure.

--Thank you for noting this error - has been corrected.

Figure 8 while the attempt to depict the connection between nosocomial infection and transmission is appreciated, it is unclear if this is the best depiction of this phenomenon. It is unclear in the text if the dots are different individuals in the green and red figures. The assumption is that because lines connect the same dots in green and red panels, however the blue grid would indicate that these are the same if transmission can be linked across the dots. If the dots are the same, then it should be clarified if these are only nosocomial cases and not any additional connected cases in the community or the percentage of the cases in the red that are community acquired vs. additional hospitalized individuals should be clarified. Is there a way in the patient-to-patient transmission to be color coded to identify dots that were green so in the hospital during infectious period and red those linked to those cases. A greater description of this figure would be appreciated by the reader to aid in understanding.

--Thank you - we have clarified the description of the figure to aid in interpretation as you have suggested, and have added additional details of the analysis in the results.

Figure 9 is this data prior to the outbreak or from supplementary immunization campaigns. Clarifying this point would help in understanding this figure.

--This reflects routine coverage assessment- the text and figure caption have been updated with this detail.

Comments on the Quality of English Language

Line 43 in the introduction should be “The three criteria include…”

--Thank you

Line 285 targeted used an extra time in the sentence.

--Thank you

Line 389 there is a period after onset that does not include the fragment of the sentence “while symptomatic”.

--Thank you

Line 459 should be “…and cross-border movement.”

--Thank you

Line 490 proactive does not need to be hyphenated.

--Thank you

Line 495 route should be root for “root cause”

--Thank you

Reviewer 2 Report

Comments and Suggestions for Authors

            The paper presents two studies of the return of measles after formal elimination, in two Asian countries: Mongolia and Cambodia. This situation of measles coming back after elimination echoes what happened earlier in some Western countries such as USA, because of importation of the virus from neighboring (or linked) countries.

The two studies are extremely well done, the paper is well documented, well presented (with excellent figures), and well written. Authors should be congratulated. The topic is important as measles eradication is technically possible (and in mind since smallpox eradication), but requires elimination in all countries and territories at the same time.

            The paper can be published as it is. Two small details could be fixed before publication (see below).

Details

1) Title of Figure 7: “Geographic distribution of confirmed measles cases - Cambodia, 2016–2017, and reported Provincial MCV1 coverage in 2015

Title is misleading: This is not “geographical coverage”, but rather time distribution… There is no MCV1 coverage in the figure.

2) Nosocomial infections seem to be a significant part of the problem. In the discussion (Lines 479-489) author could add the possibility of making strict isolation of measles cases compulsory in hospitals and clinics. This could help reducing nosocomial infections.

Author Response

Dear referee,

Thank you for the thoughtful review of our manuscript. We have fully addressed each of the identified issues below. Thanks once again.

1) Title of Figure 7: “Geographic distribution of confirmed measles cases - Cambodia, 2016–2017, and reported Provincial MCV1 coverage in 2015” Title is misleading: This is not “geographical coverage”, but rather time distribution… There is no MCV1 coverage in the figure.

--Thank you for noting this error - has been corrected.

 2) Nosocomial infections seem to be a significant part of the problem. In the discussion (Lines 479-489) author could add the possibility of making strict isolation of measles cases compulsory in hospitals and clinics. This could help reducing nosocomial infections.

--Thank you for this suggestion - we have added this text.